# Comparison of Community and Function of Dissimilatory Nitrate Reduction to Ammonium (DNRA) Bacteria in Chinese Shallow Lakes with Different Eutrophication Degrees

**Xiaowen Li [1,2], Chunlei Song [1,]\*, Zijun Zhou [3], Jian Xiao [4], Siyang Wang [5], Liu Yang [1,2], Xiuyun Cao [1] and Yiyong Zhou [1]**

1   Key Laboratory of Algal Biology, State Key Laboratory of Freshwater Ecology and Biotechnology, Institute of Hydrobiology, the Chinese Academy of Sciences, 7# Donghu South Road, Wuhan 430072, China; vayneli1221@163.com (X.L.); yangliu.os@foxmail.com (L.Y.); caoxy@ihb.ac.cn (X.C.); zhouyy@ihb.ac.cn (Y.Z.)
2   University of Chinese Academy of Sciences, Beijing 100039, China
3   Yellow River Water Resources Protection Institution, Zhengzhou 450004, China; zzj19900405@163.com
4   Guangdong Provincial Academy of Environmental Science, Guangzhou 510045, China; alexxiao811@gmail.com
5   Yellow River Institute of Hydraulic Research, Zhengzhou 450003, China; mvbctv@foxmail.com
\*   Correspondence: clsong@ihb.ac.cn; Tel.: +86-27-68780221; Fax: +86-27-6878-0709

**Abstract:** Dissimilatory nitrate reduction to ammonium (DNRA) plays an important role in controlling nitrogen (N) loading in lake ecosystems. However, studies on the linkage between DNRA bacterial community structure and lake eutrophication remain unclear. We examined the community and abundance of DNRA bacteria at six basins of four shallow lakes with different degrees of eutrophication in China. Measurements of the different forms of N and phosphorus (P) in the water column and interstitial water as well as total organic carbon (TOC) and sulfide in the sediments in summer (July 2016) were performed. The nutritional status of Lake Chaohu was more serious than that of the lakes in Wuhan, including Lake Qingling, Lake Houguan, and Lake Zhiyin by comparing geochemical and physical parameters. We found a higher abundance of the *nrfA* gene, which is a function gene of DNRA bacteria in sediments with higher contents of TOC and sulfide. Moreover, nitrate was a significant factor influencing the DNRA bacterial community structure. A significant difference of the DNRA bacterial community structure between Lake Chaohu and the lakes in Wuhan was discovered. Furthermore, DNRA bacterial abundance and community positively correlated with $NH_4^+$ and Chl *a* concentrations in Lake Chaohu, in which a percent abundance of dominant populations varied along eutrophication gradients. Overall, the abundance and community structure of the DNRA bacteria might be important regulators of eutrophication and cyanobacteria bloom in Lake Chaohu.

**Keywords:** DNRA; organic carbon; sulfide; sediment; lake; *nrfA*

## 1. Introduction

Eutrophication has become a serious environmental issue in the world [1]. Nitrogen (N) is one of the key biogenic elements affecting the eutrophication of the shallow lake ecosystem. An excessive input of N into lakes typically may cause environmental problems such as harmful algae blooms [2]. Furthermore, nitrate ($NO_3^-$) and ammonium ($NH_4^+$) are the main inorganic forms of algae utilization [3,4].

Dissimilatory $NO_3^-$ reduction to $NH_4^+$ (DNRA), a process that converts $NO_3^-$ to nitrite ($NO_2^-$) and subsequently to $NH_4^+$ [5], is a significant but previously neglected process of N cycling in freshwater environments [6]. DNRA may promote eutrophication by feeding the system with a bioavailable N ($NH_4^+$), and thus is of great ecological importance. DNRA has been reported mostly be a heterotrophic fermentation process, with an electron carbon as the electron donor, it can also be chemolithoautotrophic by using sulfide [6–10]. DNRA in sediments is affected by the interaction of various factors, mainly the concentrations of $NO_3^-$, iron, sulfide, organic carbon, temperature, and pH [11–13]. Previous studies have suggested that in the estuarine sediments, DNRA may prefer in conditions of low $NO_3^-$ concentrations and high organic electron donor availability [14]. The capacity of DNRA bacteria to cope with low $NO_3^-$ availability could be an important driver because of its higher affinity for $NO_3^-$ [15]. In the hypolimnetic sediment, organic carbon and reducing conditions shifted $NO_3^-$ reduction toward more pronounced DNRA [16]. DNRA has been estimated to account for 30% of $NO_3^-$ reduction [6], and sulfide was indicated as the key factor in some coastal systems [17]. At enough high concentrations of sulfide, DNRA can be promoted to overstep the denitrification pathway due to the inhibition of nitroso ($NO^-$) and $N_2O^-$ reductases [9]. These environmental factors change in aquatic ecosystems and thus differentially influence the structure of DNRA microbial communities. However, the relative importance of factors controlling DNRA is still unclear, and specific relationships between environmental factors and DNRA in shallow lakes needs further research.

N cycling in sediments is highly dependent on microbial processes. The bacterial community is a significant microbial regulator for DNRA in a shallow lake ecosystem. The functional *nrfA* gene encodes a pentaheme cytochrome C $NO_2^-$ reductase that catalyzes the reduction of $NO_2^-$ to $NH_4^+$ [18]. Consequently, molecular analysis of the *nrfA* gene can be used to estimate the genetic potential of DNRA in the environment [19]. It is well known that sediments' microbial communities are important to control $NO_3^-$ reduction processes [20]. An abundance of the *nrfA* gene has been found to be correlated with process rates in estuarine and coastal wetlands [21,22]. The availability of organic carbon and dominant DNRA bacteria abundance influenced the DNRA activities. Song et al. (2014) demonstrated positive relationships between the relative abundance of dominant populations and rates as well as an abundance of DNRA communities in the New River Estuary [23]. Higher levels of *nrfA* genes were consistent with a higher activity of DNRA, which both have been reported in estuary and riverine ecosystems [23,24]. Although the abundance and diversity of DNRA bacteria have been investigated in few studies, which provided a basic understanding of the microbial characteristic of the shallow lake sediments, the pathway and mechanism of organic carbon and sulfide regulating communities of DNRA bacteria as well as their functions in the process of lake eutrophication remain unclear.

In this study, different eutrophic shallow lakes such as Lake Houguan, Lake Zhiying, Lake Qingling in Wuhan, and different basins in Lake Chaohu which showed the different nutrient gradients were chosen as studied lakes. The concentrations of the different forms of N and P were determined, together with total organic carbon (TOC) and sulfide contents. The community structure and abundance of DNRA bacteria were also analyzed using the clone library and real-time quantitative PCR (qPCR) technique. The objectives of this study were (1) to compare the difference of abundance and communities of DNRA bacteria in sediments with different eutrophic level; (2) to find out the function and mechanism as well as the key regulating factor of different types of DNRA bacteria; and (3) to provide the insights into the relationship between DNRA process and eutrophication in shallow lakes.

## 2. Materials and Methods

### 2.1. Study Sites and Sample Collection

Four shallow lakes (containing Lake Qingling (QL), Lake Zhiyin (ZY), and Lake Houguan (HG), located in the Wuhan, and Lake Chaohu (CH) located in the Hefei, China) were studied in summer (10–25 July 2016) (Figure 1). Algal blooms usually occur in eutrophic lakes during summer.

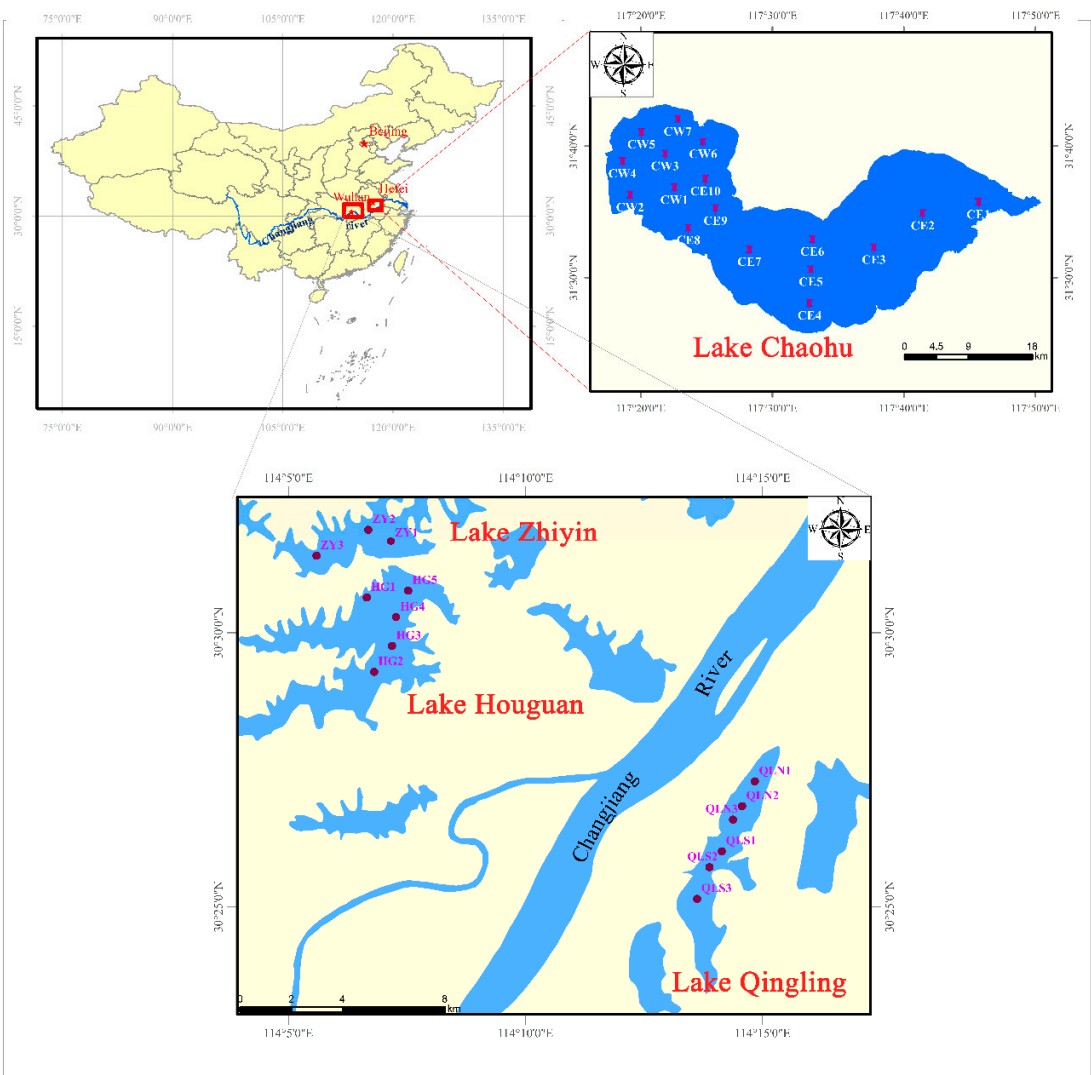

**Figure 1.** Map of studied lakes (Lake Chaohu (two basins: CHE and CHW), Lake Zhiyin (ZY), Lake Houguan (HG), and Lake Qingling (two basins: QLS and QLN)) showing the sampling sites.

Lake Qingling (30°25′–30°27′ N, 114°13′–114°14′ E) is about 9.6 km$^2$. The average depth is 1.7 m. It was artificially divided into two basins. The south basin (QLS) is covered by hydrophytes. In contrast, the north basin (QLN) used to be a fish farm, leading to the disappearance of hydrophytes [25]. There were six sample sites, three in the QLN (QLN1–QLN3) and the other in the QLS (QLS1–QLS3). Lake Zhiyin (30°52′–30°55′ N, 114°09′–114°14′ E) is approximately 30 km$^2$. The average depth is 1.5 m. The cultivation density of ZY is reasonable, and the land-use mode is mainly residential area, fish pond, and cultivated land. Three sample sites were established (ZY1–ZY3). Lake Houguan (30°47′–30°52′ N, 114°08′–114°15′ E) is about 34.4 km$^2$. The average depth is 1.6 m. Its aquaculture density is reasonable; some of the areas show submerged plants. Five sample sites were set (HG1–HG5). Lake Chaohu (31°25′–31°43′ N, 117°16′–117°51′ E), the fifth largest shallow freshwater lake in China, is located in Anhui province. The largest water area is about 825 km$^2$ and the depth is about 2 m. According to the nutritional status, CH was divided into two basins. The west basin (CHW) has a large population density and relatively serious pollution, and is undergoing eutrophication. On the contrary, the water quality in the east basin (CHE) is undergoing moderate eutrophication. Seventeen sampling sites were set, with eight in the CHE (CHE1–CHE8) and nine in the CHW (CHW1–CHW9).

Water samples (0–50 cm) were taken by an organic glass hydrophore. Surface sediment samples (0–10 cm) were collected using a Pedersen mud collector. Sediment samples for DNA extraction and

molecular analysis were frozen at −80 °C. Interstitial water samples were extracted from sediment by centrifugation at 4000 rpm for 15 min.

### 2.2. Chemical Analysis

Physiochemical parameters including the water temperature (T), pH, and dissolved oxygen (DO) of surface water samples were measured in situ using a YSI meter. The transparency (SD) was measured by a Secchi disk. Surface water samples were filtered through the GF/C filter for Chlorophyll *a* (Chl *a*) measurement, and then analyzed based on the ethanol extraction method [26]. The surface and interstitial water for nutrient analysis were filtered through a 0.45 μm cellulose acetate membrane. Soluble reactive phosphorus (SRP) measurement was determined following the molybdate blue method [27]. Total phosphorus (TP) and dissolved total phosphorus (DTP) were determined according to the digestion method [28]. Analyses of different forms of N (TN, dissolved total nitrogen, or DTN, $NH_4^+$, and $NO_3^-$) followed the method reported by the Standard Methods [29]. Total organic carbon (TOC) in the sediment was determined by a TOC Analyzer (multi N/C 3100, Analytikjena, Germany) following the manufacturer's instructions. Acid volatile sulfide (AVS) was analyzed based on methylene blue spectrophotometry [30].

### 2.3. DNA Extraction and Quantitative PCR

Sediment DNA was extracted using the Power Soil DNA kit (Mo Bio Laboratories, Carlsbad, CA, USA) following the manufacturer's instructions. The abundance of DNRA bacteria targeting *nrfA* gene were determined by real-time quantitative polymerase chain reaction (qPCR) and using an ABI STEPONE PLUS thermo cycler (Applied Biosystems, Foster City, CA, USA). The primers for the *nrfA* gene are nrfA2aw/nrfAR1 [19,31]. Triplicate reactions were performed with a SYBR Green PCR Master Mix, and the reaction system (20 μL) contained 10 μL of 2 × Mix, 0.4 μM of each primer and 1 μL of DNA. qPCR was conducted in an Applied Biosystems 7900 Real-Time PCR System (Applied Biosystems, USA), and the qPCR conditions were as follows: an initial 5 min at 94 °C, 40 cycles of 94 °C for 30 s, 52 °C for 45 s, and 72 °C for 20 s, followed by 5 min at 72 °C. The PCR products were confirmed by melt curve analysis and agarose gel electrophoresis. A known copy number of linearized plasmids of the *nrfA* gene clone was used as the standard for DNRA bacteria qPCR. The amplification efficiencies for the *nrfA* gene were 90% ($R^2 > 0.98$).

The *nrfA* gene sequences for DNRA bacteria are under accession numbers MN165039-MN165060.

### 2.4. Cloning, Sequencing, and Phylogenetic Analyses

Amplifications of the *nrfA* gene for cloning and sequencing were performed. The cloning was carried out using the pGEM T-Easy vector (Promega) system according to the instructions of the manufacturer. Positive clones were randomly selected and sent to the Magigene Company (Guangzhou, China) for sequencing. The obtained sequences were deposited into the GenBank database. The retrieved sequences and reference sequence were aligned using CLUSTAL X, and the phylogenetic neighbor-joining trees were constructed using MEGA version 5.1.

### 2.5. Statistical Analysis

Pearson's correlation coefficient analyses were carried out using SPSS 20.0 (Chicago, IL, USA) to test the relationships between parameters. Redundancy analysis (RDA) was used to examine the relationships between biological and environmental variables. A heatmap run with R software was performed to study the differences of the community structure among the four lakes and diversities and abundances of DNRA bacteria. Operational taxonomic units (OTUs) clustering were performed by MOTHUR software [32].

## 3. Results

### 3.1. Nutrients Level

There was little variation in pH, T, and DO among the surface water samples of the studied lakes. In detail, the average of T, pH, and DO in Lake Chaohu and lakes in Wuhan were about 29.5 °C, 8.26, and 7.38 mg $L^{-1}$, respectively. The SD of Lake Chaohu (0.05–0.20 m) was significantly lower than that of lakes in Wuhan (0.25–0.40 m) ($p < 0.01$). Significantly higher concentrations of TN, TP, $NO_3^-$, and $NH_4^+$ in the water column of Lake Chaohu west basin were recorded. Additionally, the nutrient state of Lake Chaohu showed spatial heterogeneity ($p < 0.05$), which increased gradually from the east basin to the west (Figure 2). In addition, the Chl *a* concentration in the Lake Chaohu west basin was the highest. Cyanobacteria bloom occurred in the west basin while sampling. On the other hand, the Chl *a* concentration in lakes in Wuhan were generally low, and did not form algae bloom except for the north basin of Lake Qingling (Figure 2c). In Lake Qingling, different from the south basin, which was dominated by hydrophytes, significantly higher concentrations of N, P, and Chl *a* were recorded in the north basin dominated by algae (Figure 2). The concentration of TP in the water column of Lake Chaohu and the north basin of Lake Qingling showed significantly ($p < 0.05$) higher levels than those of the other lakes (Figure 2d). Moreover, the DTP concentration in the north basin of Lake Qingling was also significantly ($p < 0.05$) higher (Figure 2e). Meanwhile, TN and DTN concentrations in the water column of Lake Chaohu west basin exhibited a higher value (Figure 2a,b). A different distribution pattern of $NO_3^-$ concentrations in the column water and interstitial water was found. A significantly higher level of $NO_3^-$ concentrations in the water column of Lake Chaohu and the interstitial water of the lakes in Wuhan was recorded (Figure 2g,j). In general, concentrations of $NH_4^+$ in the water column and interstitial water of Lake Chaohu were significantly ($p < 0.05$) higher than that of lakes in Wuhan, especially in the west basin of Lake Chaohu (Figure 2h,k). The $NH_4^+$ concentration in interstitial water was higher than that of the water column, which indicated that $NH_4^+$ may tend to release upward. SRP concentration did not show a significant difference in both the water column and interstitial water in all the studied lakes, except for the north basin of Lake Qingling, which had an especially high value (Figure 2i,l). The contents of TOC in the sediments of Lake Chaohu was remarkably ($p < 0.05$) lower than those of lakes in Wuhan (Figure 3a). The AVS contents in the sediments of Lake Zhiyin and the south basin of Lake Qingling were higher ($p < 0.01$) at a low level in sediments than in the other lakes (Figure 3b). Due to hydrophytes, both the TOC and AVS contents in the Lake Qingling south basin was significantly exceeded by those of the other lakes. The decomposition of organic matter consumes large amounts of oxygen and causes a reductive condition, which also increases the AVS content [33].

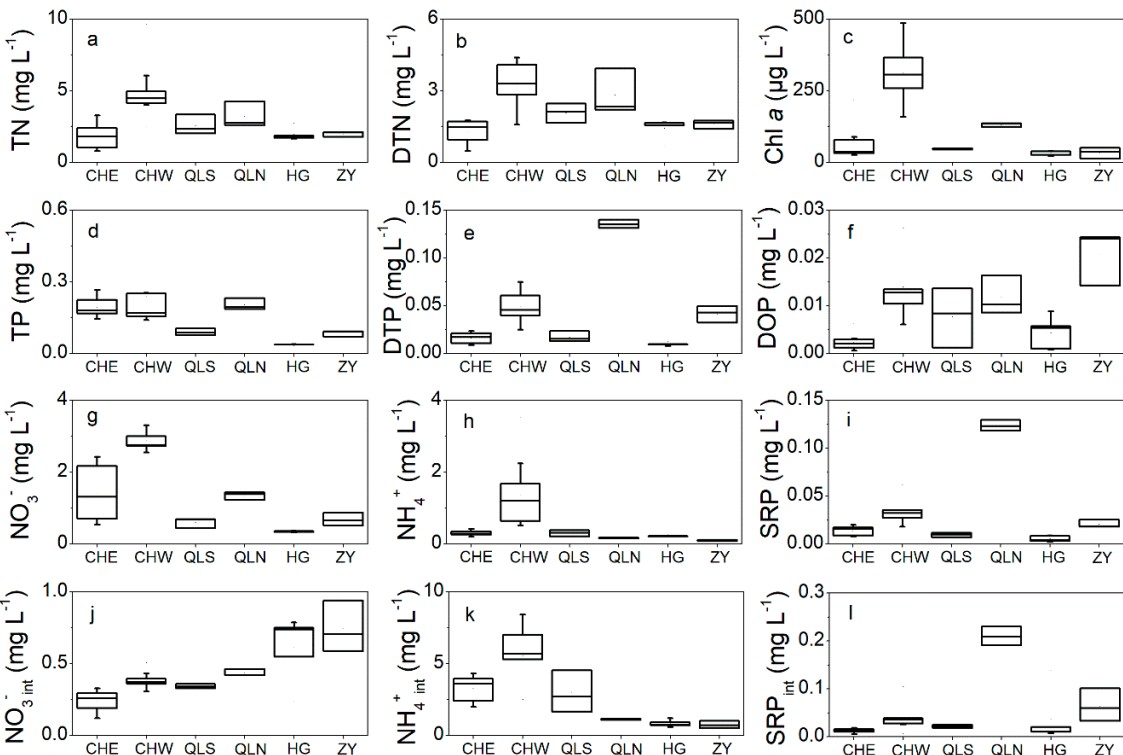

**Figure 2.** Boxplots showed the mean of chemical parameters of water column and interstitial water in the east (CHE) and west basins (CHW) of Lake Chaohu, south (QLS) and north (QLN) basins of Lake Qingling, Lake Houguan (HG), and Lake Zhiyin (ZY). Horizontal lines indicate the median, small squares show the mean, asterisks indicate the outliers, the boxes give the 25th and 75th percentiles, and bars show the range from the 5th to 95th percentiles. (**a**) TN, total nitrogen; (**b**) DTN, dissolved total nitrogen; (**c**) Chl *a*; (**d**) TP, total phosphorus; (**e**) DTP, dissolved total phosphorus; (**f**) DOP, dissolved organic phosphorus; (**g**) $NO_3^-$; (**h**) $NH_4^+$; (**i**) SRP, soluble reactive phosphorus; (**j**) $NO_3^-{}_{int}$, nitrate in interstitial water; (**k**) $NH_4^+{}_{int}$, ammonium in interstitial water; (**l**) $SRP_{int}$, soluble reactive phosphorus in interstitial water).

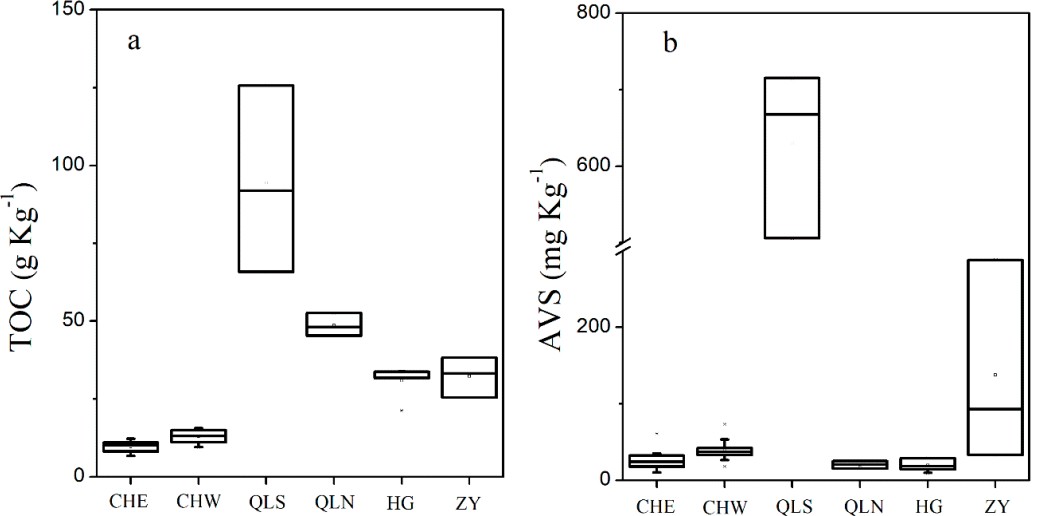

**Figure 3.** Boxplots showed the mean contents of total organic carbon (TOC) (**a**) and acid volatile sulfide (AVS) (**b**) in sediments of the east (CHE) and west basins (CHW) of Lake Chaohu, south (QLS) and north (QLN) basins of Lake Qingling, Lake Houguan (HG), and Lake Zhiyin (ZY). Horizontal lines indicate the median, small squares show the mean, asterisks indicate the outliers, the boxes give the 25th and 75th percentiles, and bars show the range from the 5th to 95th percentiles.

### 3.2. Abundance and Community Structure of Bacteria Mediating DNRA Process

Variations in the functional gene copy numbers showed significantly differences between Lake Chaohu and lakes in Wuhan. In detail, an abundance of the *nrfA* gene in Lake Chaohu ranged from $3.12 \times 10^9$ to $9.06 \times 10^9$, while it ranged from $4.09 \times 10^9$ to $1.96 \times 10^{10}$ copies $g^{-1}$ dry sediment in the lakes in Wuhan (Figure 4). As a result, the *nrfA* gene abundances of the lakes in Wuhan were significantly ($p < 0.01$) higher than that of Lake Chaohu.

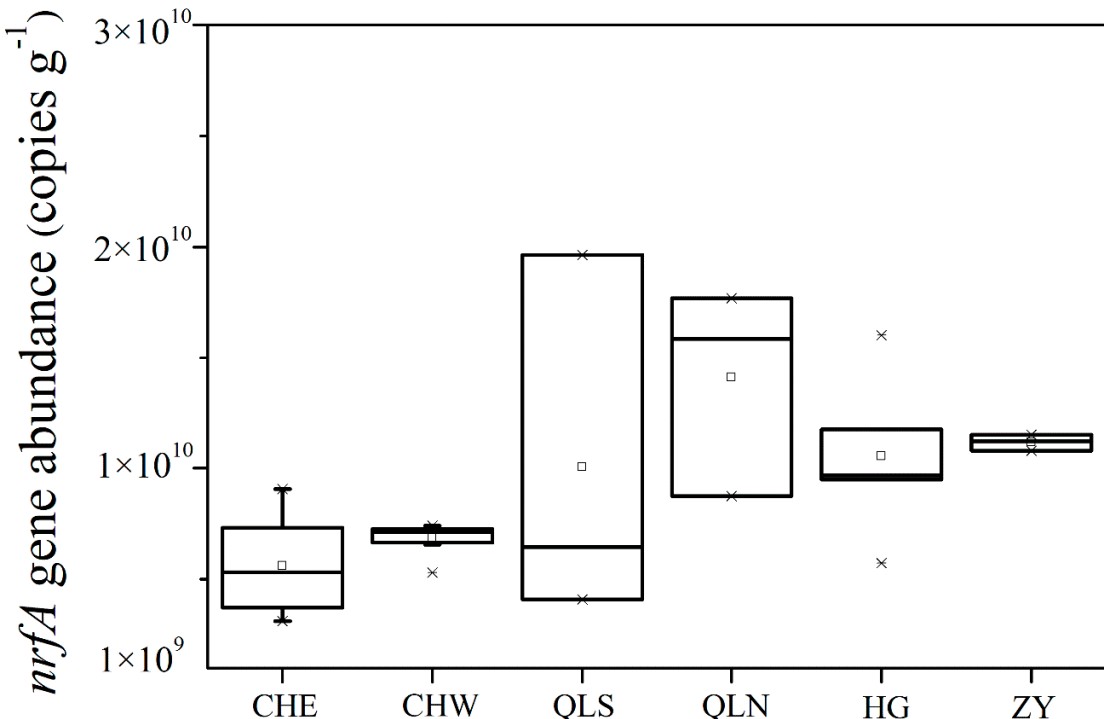

**Figure 4.** Boxplots showed the abundance of the *nrfA* gene in the sediments of the east (CHE) and west basins (CHW) of Lake Chaohu, south (QLS) and north (QLN) basins of Lake Qingling, Lake Houguan (HG), and Lake Zhiyin (ZY). Horizontal lines indicate the median, small squares show the mean, asterisks indicate the outliers, the boxes give the 25th and 75th percentiles, and bars show the range from the 5th to 95th percentiles.

A total of 650 *nrfA* gene clones were clustered into 22 OTUs on the basis of a 15% cutoff, which could be grouped into three clusters. Clusters among different lakes were conducted based on the similarities of the OTUs' relative abundances, which were shown with heatmaps to illustrate the difference of the DNRA bacterial community structures in different lakes. The phylogenetic tree of the *nrfA* protein showed that all the sequences were connected to estuarine sediments, agricultural soil, and pure culture strains of the genera *Desulfitobacterium*, *Desulfosporosinus*, *Desulfitobacterium*, *Geobacter*, *Anaerolinea*, *Anaeromyxobacter*, *Verrucomicrobiae*, *Helicobacter*, *Porphyromonas*, and *Bcteroides* (Figure 5). Our findings revealed the high diversity of bacteria that participate in the DNRA process in the studied lakes sediments. DNRA bacteria in sediments of Lake Chaohu could be clustered into one group with OTU1 as the dominant OTU, which was affiliated to *Desulfitobacterium*. However, in the sediments of lakes in Wuhan, DNRA bacteria could also grouped together with OTU2 as the dominant OTU, which was affiliated to *Bacteroides*.

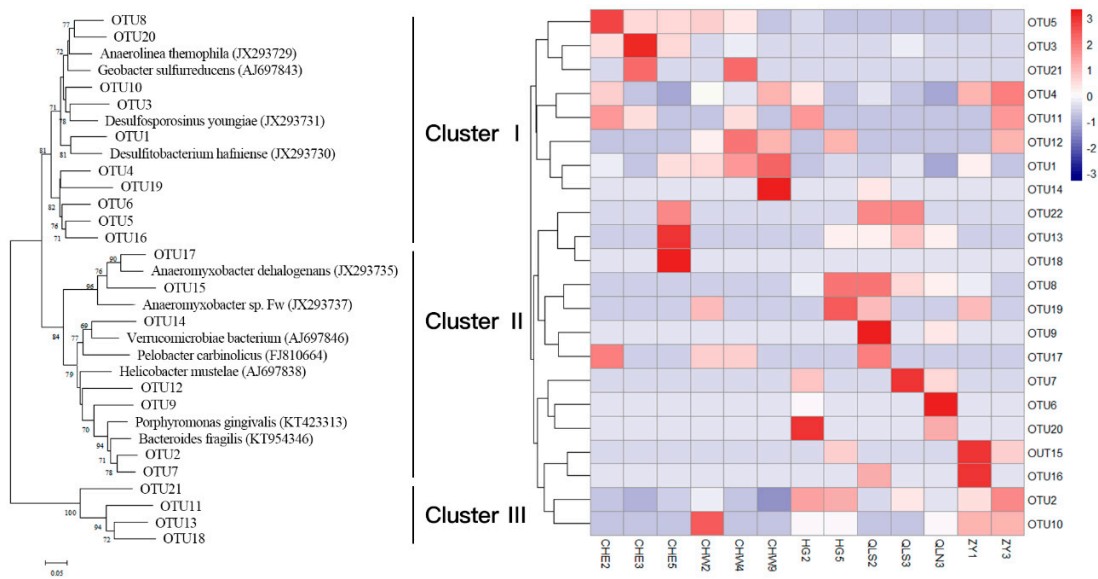

**Figure 5.** Neighbor-joining phylogenetic tree of *nrfA* sequences and heatmap showing clustering among different lakes based on operational taxonomic units' (OTUs) relative abundance.

### 3.3. Relationship Between Abundance, Community Structure of DNRA Bacteria, and Nutrients

The abundance and diversity of DNRA bacteria varied among the studied lakes with different trophic states. Significantly positive relationships existed between *nrfA* gene abundance and DTP ($p < 0.05$), SRP ($p < 0.05$), $NO_3^-$ ($p < 0.01$), and Chl *a* ($p < 0.01$) concentrations in the water column of Lake Chaohu. Negative correlation with SD and *nrfA* gene abundance was recorded. In addition, *nrfA* gene abundance was significantly positively related to $NO_3^-$ ($p < 0.01$) and $NH_4^+$ ($p < 0.01$) concentrations in interstitial water as well as contents of TOC and AVS in sediments ($p < 0.05$) (Table 1). However, in lakes located in Wuhan, no relationship between *nrfA* gene abundance and nutrient parameters was found (Table 2). Chl *a* concentration of Lake Chaohu was influenced by not only various forms of N and P in the water column and interstitial water but also TOC and AVS contents. Moreover, concentrations of $NH_4^+$ in the water column and $NO_3^-$ in interstitial water played an important role in Chl *a*. A positive correlation between $NH_4^+$ concentration in the water column with $NO_3^-$ concentration in the interstitial water and TOC ($p < 0.01$) and AVS ($p < 0.05$) contents in sediments were observed; all of these factors controlled the abundance of the *nrfA* gene.

Additionally, RDA analysis showed that the dominant OTU (OTU1) in the sediment of Lake Chaohu was regulated positively by not only the concentrations of $NO_3^-$ and $NH_4^+$ in the water column, but also by $NO_3^-$ concentrations in interstitial water, although these parameters were negatively correlated with OTU2. Chl *a* concentration and *nrfA* gene abundance were also positively related with OTU1 and OTU2 (Figure 6a). On the other hand, the OTU2 prevailing in sediments of lakes in Wuhan was mainly positively influenced by concentrations of $NO_3^-$ in interstitial water, while it was negatively correlated with TN and Chl *a* concentrations in the water column and $NH_4^+$ in interstitial water (Figure 6b). The SRP concentration showed less effect on the DNRA bacteria community structure of lakes in Wuhan.

**Table 1.** Pearson's correlation coefficients among different parameters in Lake Chaohu.

| □ | *nrfA* | TN | DTN | TP | DTP | DOP | SRP | Chl *a* | $NH_4^+$ | $NO_3^-$ | $SRP_{int}$ | $NH_4^+{}_{int}$ | $NO_3^-{}_{int}$ | AVS | TOC | T | DO | pH |
|---|---|---|---|---|---|---|---|---|---|---|---|---|---|---|---|---|---|---|
| *nrfA* | 1 | | | | | | | | | | | | | | | | | |
| TN | 0.16 | 1 | | | | | | | | | | | | | | | | |
| DTN | 0.20 | 0.92 ** | 1 | | | | | | | | | | | | | | | |
| TP | −0.14 | 0.47 | 0.30 | 1 | | | | | | | | | | | | | | |
| DTP | 0.53 * | 0.51 * | 0.48 * | −0.16 | 1 | | | | | | | | | | | | | |
| DOP | 0.31 | 0.56 * | 0.51 * | −0.02 | 0.89 ** | 1 | | | | | | | | | | | | |
| SRP | 0.57 * | 0.47 | 0.46 | −0.20 | 0.97 ** | 0.79 ** | 1 | | | | | | | | | | | |
| Chl *a* | 0.66 ** | 0.66 ** | 0.67 ** | −0.04 | 0.79 ** | 0.73 ** | 0.77 ** | 1 | | | | | | | | | | |
| $NH_4^+$ | 0.29 | 0.62 ** | 0.58 * | −0.00 | 0.79 ** | 0.77 ** | 0.77 ** | 0.77 ** | 1 | | | | | | | | | |
| $NO_3^-$ | 0.63 ** | 0.65 ** | 0.71 ** | −0.08 | 0.64 ** | 0.55 * | 0.63 ** | 0.81 ** | 0.51 * | 1 | | | | | | | | |
| $SRP_{int}$ | 0.42 | 0.59 * | 0.65 ** | 0.08 | 0.62 ** | 0.49 * | 0.67 ** | 0.65 ** | 0.81 ** | 0.53 * | 1 | | | | | | | |
| $NH_4^+{}_{int}$ | 0.63 ** | 0.32 | 0.35 | −0.03 | 0.58 * | 0.64 ** | 0.53 * | 0.64 ** | 0.45 | 0.47 | 0.24 | 1 | | | | | | |
| $NO_3^-{}_{int}$ | 0.65 ** | 0.46 | 0.43 | −0.04 | 0.83 ** | 0.82 ** | 0.79 ** | 0.79 ** | 0.70 ** | 0.64 ** | 0.65 ** | 0.62 ** | 1 | | | | | |
| AVS | 0.53 * | 0.27 | 0.12 | −0.09 | 0.50 * | 0.42 | 0.49 * | 0.65 ** | 0.52 * | 0.29 | 0.18 | 0.38 | 0.36 | 1 | | | | |
| TOC | 0.57 * | 0.47 | 0.41 | 0.03 | 0.48 | 0.48 | 0.43 | 0.56 * | 0.62 ** | 0.34 | 0.37 | 0.43 | 0.26 | 0.59 * | 1 | | | |
| T | 0.20 | 0.53 * | 0.69 ** | 0.09 | 0.31 | 0.44 | 0.26 | 0.52 * | 0.44 | 0.41 | 0.49 * | 0.42 | 0.42 | −0.04 | 0.35 | 1 | | |
| DO | 0.13 | 0.26 | 0.38 | 0.29 | 0.11 | 0.17 | 0.12 | 0.30 | 0.45 | 0.11 | 0.60 * | 0.33 | 0.37 | −0.01 | 0.13 | 0.51 * | 1 | |
| pH | 0.17 | 0.14 | 0.27 | 0.33 | 0.06 | 0.12 | 0.06 | 0.18 | 0.30 | 0.01 | 0.41 | 0.43 | 0.31 | −0.13 | 0.10 | 0.52 * | 0.91 ** | 1 |

*nrfA*: *nrfA* gene abundance; TN: total nitrogen; DTN: dissolved total nitrogen; TP: total phosphorus; DTP: dissolved total phosphorus; DOP: dissolved organic phosphorus; SRP: soluble reactive phosphorus; Chl *a*: chlorophyll *a*; $SRP_{int}$: soluble reactive phosphorus in interstitial water; $NH_4^+{}_{int}$: ammonium in interstitial water; $NO_3^-{}_{int}$: nitrate in interstitial water; AVS: acid volatile sulfide; TOC: total organic carbon; T: temperature; DO: dissolved oxygen. Significance at ** $\alpha = 0.01$ level; * $\alpha = 0.05$ level.

**Table 2.** Pearson's correlation coefficients among different parameters in Lake Qingling, Lake Zhiyin, and Lake Houguan.

| □ | nrfA | TN | DTN | TP | DTP | DOP | SRP | Chl a | NH$_4^+$ | NO$_3^-$ | SRP$_{int}$ | NH$_4^+{}_{int}$ | NO$_3^-{}_{int}$ | AVS | TOC | T | DO | pH |
|---|---|---|---|---|---|---|---|---|---|---|---|---|---|---|---|---|---|---|
| nrfA | 1 | | | | | | | | | | | | | | | | | |
| TN | 0.04 | 1 | | | | | | | | | | | | | | | | |
| DTN | 0.05 | 0.83 ** | 1 | | | | | | | | | | | | | | | |
| TP | 0.22 | 0.61 * | 0.67 ** | 1 | | | | | | | | | | | | | | |
| DTP | 0.29 | 0.61 * | 0.63 * | 0.89 ** | 1 | | | | | | | | | | | | | |
| DOP | 0.37 | 0.35 | 0.19 | 0.44 | 0.61 * | 1 | | | | | | | | | | | | |
| SRP | 0.19 | 0.58 * | 0.68 ** | 0.89 ** | 0.95 ** | 0.35 | 1 | | | | | | | | | | | |
| Chl a | 0.20 | 0.71 ** | 0.72 ** | 0.69 ** | 0.68 ** | 0.09 | 0.76 ** | 1 | | | | | | | | | | |
| NH$_4^+$ | −0.34 | 0.12 | 0.03 | −0.24 | −0.52 | −0.62 * | −0.42 | 0.05 | 1 | | | | | | | | | |
| NO$_3^-$ | 0.28 | 0.65 * | 0.65 * | 0.93 ** | 0.92 ** | 0.58 * | 0.87 ** | 0.69 ** | −0.34 | 1 | | | | | | | | |
| SRP$_{int}$ | 0.19 | 0.59 * | 0.69 ** | 0.88 ** | 0.94 ** | 0.34 | 0.99 ** | 0.76 ** | −0.41 | 0.86 ** | 1 | | | | | | | |
| NH$_4^+{}_{int}$ | −0.35 | 0.54 * | 0.38 | 0.31 | 0.01 | 0.02 | −0.01 | 0.22 | 0.62 * | 0.24 | −0.01 | 1 | | | | | | |
| NO$_3^-{}_{int}$ | 0.11 | −0.32 | −0.45 | −0.21 | −0.08 | 0.26 | −0.20 | −0.41 | −0.40 | −0.13 | −0.21 | −0.38 | 1 | | | | | |
| AVS | −0.27 | 0.09 | 0.07 | 0.16 | −0.07 | 0.19 | −0.12 | −0.11 | 0.12 | 0.00 | −0.14 | 0.62 * | −0.25 | 1 | | | | |
| TOC | −0.35 | 0.57 * | 0.48 | 0.45 | 0.14 | −0.02 | 0.19 | 0.34 | 0.51 | 0.35 | 0.19 | 0.93 ** | −0.54 * | 0.63 * | 1 | | | |
| T | −0.05 | −0.47 | −0.25 | −0.23 | −0.01 | −0.15 | 0.06 | −0.10 | −0.53 * | −0.18 | 0.06 | −0.75 ** | 0.49 | −0.46 | −0.69 ** | 1 | | |
| DO | −0.40 | −0.01 | −0.04 | −0.21 | −0.21 | −0.26 | −0.12 | 0.19 | 0.06 | −0.23 | −0.11 | 0.03 | 0.00 | 0.23 | 0.06 | 0.38 | 1 | |
| pH | −0.23 | −0.37 | −0.49 | −0.40 | −0.26 | −0.24 | −0.23 | −0.08 | −0.13 | −0.41 | −0.22 | −0.42 | 0.44 | −0.18 | −0.45 | 0.65 * | 0.74 ** | 1 |

nrfA: nrfA gene abundance; TN: total nitrogen; DTN: dissolved total nitrogen; TP: total phosphorus; DTP: dissolved total phosphorus; DOP: dissolved organic phosphorus; SRP: soluble reactive phosphorus; Chl a: chlorophyll a; SRP$_{int}$: soluble reactive phosphorus in interstitial water; NH$_4^+{}_{int}$: ammonium in interstitial water; NO$_3^-{}_{int}$: nitrate in interstitial water; AVS: acid volatile sulfide; TOC: total organic carbon; T: temperature; DO: dissolved oxygen. Significance at ** $\alpha$ = 0.01 level; * $\alpha$ = 0.05 level.

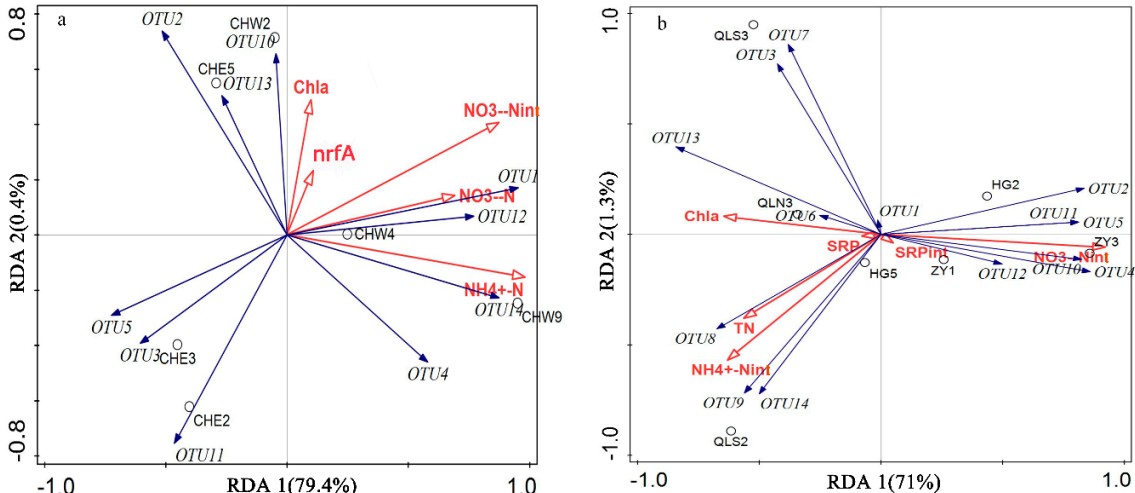

**Figure 6.** Redundancy analysis (RDA) deciphering the relationship between operational taxonomic units (OTUs) and environmental variables in Lake Chaohu (**a**), Lake Qingling, Lake Houguan, and Lake Zhiyin (**b**). Red arrows indicate the environmental factors, and blue arrows indicate the OTUs. Circles represent the sampling sites.

## 4. Discussion

In this study, we reported the abundance and community structure distributions of DNRA bacteria in studied lakes. A distinct trophic gradient status was put in evidence among Lake Chaohu, Lake Qingling, Lake Zhiyin, and Lake Houguan in Wuhan. The abundances of the *nrfA* gene measured in our study were similar to those researches reported in the eutrophic New River Estuary, Lake Chaohu, and Yellow River Estuary, which used the same primers [23,34,35]. Compared to previous studies, the functional gene abundance of DNRA was higher than that of denitrification ($1.25 \times 10^6$ to $9.44 \times 10^8$ copies g$^{-1}$), nitrification ($6.91 \times 10^6$ to $1.7 \times 10^9$ copies g$^{-1}$), and anaerobic ammonia oxidation ($3.74 \times 10^6$ to $9.48 \times 10^6$ copies g$^{-1}$) among the same lakes according to the qPCR technique [34,36]. The *nrfA* gene exists in different groups of bacteria including *Proteobacteria*, *Planctomycetes*, *Bacteroides*, and *Firmicutes* [19]. Furthermore, respiratory metabolism pathways in DNRA bacteria are varying, including fermentation, denitrification, anammox, and sulfate reduction [37,38]. Thus, the diversity and abundance of DNRA bacteria in sediments might be greater than other organisms that take part in N cycling because of metabolic versatility [23].

In Lake Chaohu, *nrfA* gene abundance was positively regulated by various nutrients' factors, including NO$_3^-$ concentrations in the water column and interstitial water as well as TOC and AVS contents in sediments (Table 1), which also showed significant spatial heterogeneity along the nutrient status (Figure 4). However, no relationship between *nrfA* gene abundance and nutrients was observed in the lakes of Wuhan (Table 2). Higher *nrfA* gene abundances were measured in the lakes of Wuhan sites where AVS and TOC contents in sediments were rich. It has been demonstrated that the quantity and availability of organic carbon strongly affect DNRA bacterial abundance [24,39]. As such, it is possible that these factors resulted in significantly lower *nrfA* gene abundance in the sediments of Lake Chaohu. Similar studies have been documented in the eutrophic New River Estuary and Yellow River Estuary [23,35]. Therefore, organic carbon might greatly limit DNRA bacterial abundance in Lake Chaohu. On the other hand, without available carbon, some sulfide and metals also can provide viable alternative electron donors to promote DNRA [40,41]. This mechanism may be crucial in the Upper Floridan Aquifer, which is a low-carbon system [42]. It is similar to Lake Chaohu, in which the dominant OTUs were closely related with *D. hafniense*, which can utilize organic carbon and S$_2^-$ as electron sources [43]. Hence, we speculated that the DNRA bacteria in Lake Chaohu make more use of sulfide.

The highest *nrfA* gene abundance was detected in the south basin of Lake Qingling, where $NO_3^-$ levels were low in interstitial water and there were high TOC contents in sediment. Besides, higher C/N ratios were also observed in the sediments of lakes in Wuhan. DNRA bacteria have a higher affinity for $NO_3^-$ and favor carbon-rich and $NO_3^-$-limited environments [15,44]. In agreement with our results, previous works had shown *nrfA* abundance to be higher in conditions of high C/N ratios of lakes sediments with different nutritional states [22,45]. Therefore, C/N might also be one of the reasons for the differences in DNRA bacterial abundance between Lake Chaohu and the lakes in Wuhan.

In the present study, we investigated the major bacterial groups involved in DNRA. OTUs associated with sediments in Lake Chaohu and lakes in Wuhan-related sequences were affiliated with *Desulfitobacterium* and *Bacteroides*, respectively. All these microbes are known to carry *nrfA* [31]. As shown in the heatmap (Figure 5), the relative abundance of DNRA bacteria varied between Lake Chaohu and the lakes in Wuhan. The microbial community present in the sediment of Lake Chaohu was strongly influenced by $NO_3^-$ in the water column and interstitial water, $NH_4^+$ and Chl *a* concentrations in the water column, as well as *nrfA* gene abundance. On the other hand, concentrations of $NO_3^-$ and $NH_4^+$ in interstitial water and Chl *a* also affected the communities of DNRA bacteria of lakes in Wuhan (Figure 6). Many studies have indicated that the abundance and diversity of DNRA bacteria in sediments might be affected by many factors, including the concentration gradient of $NO_3^-$, sulfide, organic carbon, and DO [11–13]. In the Yellow River Estuary, the diversity of the DNRA community also showed a positive and significant correlation with $NO_3^-$ concentration in the bottom water [35]. Consequently, $NO_3^-$ not only could affect the abundance of DNRA bacteria, but it could also be an important environmental factor controlling the DNRA community, highlighting the tight coupling of DNRA with the substrate.

In Lake Chaohu, as in most of the aquatic ecosystems, the N cycle is predominantly mediated by microbe processes. A number of authors also have indicated a positive and significant linkage between DNRA activity and *nrfA* gene abundance [21–24,46]. Values of TN, DTN, TP, $NO_3^-$, $NH_4^+$, and Chl *a* tended to be higher in the west basin. Notably, cyanobacteria bloom formed in Lake Chaohu seriously every year [47]. Moreover, cyanobacteria bloom occurred in the west basin while sampling, and the Chl *a* concentration was remarkable (Figure 2c). It is generally acknowledged that cyanobacteria prefer $NH_4^+$ due to its higher absorption affinity and faster absorption rates [4,48]. In our investigation, Chl *a* concentration was positively correlated with $NH_4^+$ concentrations in the water column and interstitial water ($p < 0.01$) (Table 1). Furthermore, positive correlations between *nrfA* gene abundance and $NH_4^+$ in the water column and interstitial water as well as Chl *a* concentrations indicated that DNRA may be an important process for sediment $NH_4^+$ flux and the maintenance of cyanobacteria bloom. In fact, the abundance of the *nrfA* gene in the north basin of Lake Qingling was higher than that of the south, which partly explained the contribution of DNRA bacteria to algal biomass.

The composition and diversity of functional bacteria can be another microbial regulator of DNRA in shallow lake sediments. Song demonstrated that the relative abundance of dominant populations was positively correlated with the activity and abundance of DNRA communities in the New River Estuary [23], which could support our results for Lake Chaohu. The proportion of OTU1 increased with the nutritional status, which was higher in the west basin (Figure 5). Furthermore, the community of DNRA bacteria in Lake Chaohu was also related to Chl *a*. These findings emphasized the importance of the dominant population and their roles in the overall community activities. Finally, changes in activities could reflected by the abundance and community structure of DNRA bacteria, supporting a linkage between the structure and function of microbial communities. Therefore, DNRA might be considered as an important supplementary pathway of $NH_4^+$ production in Lake Chaohu during the cyanobacteria bloom period.

## 5. Conclusions

This study reveals that variability in the abundance and community structure of DNRA bacteria was largely driven by environmental factors. The abundance of bacteria capable of DNRA was mainly

regulated by organic carbon, sulfide, $NO_3^-$, and C/N. The composition and diversity of DNRA bacteria was more affected by $NO_3^-$. Moreover, the relative abundance of dominant populations varied along the eutrophication gradient. These factors were the reasons for the differences in the DNRA bacterial community structure between Lake Chaohu and the lakes in Wuhan. Additionally, positive relationships between the abundance and community composition of DNRA bacteria and concentrations of $NH_4^+$ and Chl *a* during the cyanobacterial bloom in Lake Chaohu suggested that DNRA may be an important process for sediment $NH_4^+$ flux and the maintenance of cyanobacteria bloom. Our findings provide new insights into the characteristics and function of DNRA communities in shallow lake ecosystems.

**Author Contributions:** Conceptualization, X.L. and C.S.; methodology, X.L.; software, X.L. and Z.Z.; validation, X.L., J.X., and Z.Z.; investigation, X.L., L.Y., and S.W.; data curation, X.L.; writing—original draft preparation, X.L.; writing—review and editing, C.S.; supervision, X.C.; project administration, Y.Z.; funding acquisition, Y.Z. and C.S. All authors have read and agreed to the published version of the manuscript.

**Funding:** This research was funded by the National Natural Science Foundation of China (41877381; 41573110), National Key Research and Development Program of China (2016YFE0202100), the Major Science and Technology Program for Water Pollution Control and Treatment (2017ZX07603) and State Key Laboratory of Freshwater Ecology and Biotechnology (2019FBZ01).

**Acknowledgments:** We thank Z.Z. and S.W. for their guidance on experiments.

**Conflicts of Interest:** The authors declare no conflict of interest.

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
