# Peer review of "Comparison of Community and Function of Dissimilatory Nitrate Reduction to Ammonium (DNRA) Bacteria in Chinese Shallow Lakes with Different Eutrophication Degrees"

_water, doi:10.3390/w12010174_

Round 1
Reviewer 1 Report
Review of ' Comparison of Community and functions……'
Xiaowen et al., 2019
Reviewed Dec. 17, 2019
This paper presents an extensive data set relating the bacterial communities and environmental conditions in four Chinese lakes with varying degrees of eutrophication. An important finding is the relationship between NO3, NH4, DNRA bacteria and correlation with cyanobacteria outbreaks in one of the lakes (CHW). Although, I’m not an expert in microbiology, this appears to be a significant finding and this, along with the other correlations presented particularly in Tables 2 and 3, I feel, makes this study worthy of publication. I have included a number of editing comments below, aimed at improving the readability of the manuscript;
1) Table 1; The caption is incomplete. Which lake is which? Are these surface readings, or at depth, or interstitial water? Were they taken at different locations on the same day or different times at the same location? This information needs to be included in the caption to allow the figure to ‘stand alone’ without the necessity of referring to the text.
2) Table 1; Generally too many significant digits reported; e.g. report DO to only one decimal and don’t report decimals for ORP. Please check ORP values. They appear too low (<200 mV) for aerobic water. Perhaps forgot to add the probe constant (~200 mV) to the field reading.
3) Table 1; All of the readings (3-8) from each lake are generally similar. Thus, there is no need to report all these numbers. Simply report the mean +/- sd and n for each lake to save space.
4) With the space saved by condensing Table 1, double the size of Fig. 2. This figure is an important figure but it is essentially illegible because the font size is much too small, etc. Decrease the number of values on the Y axes and increase their font size. e.g. list TN values at 0, 5 and 10 only, not 0, 2, 4, 6, 8, 10. Simplify the Y values on all the plots similarly. This is particularly important because you have used different Y scales for each of the N and P parameters, which at first glance, makes the values look similar when actually they are much different. The reader needs to see immediately that the scales are different (i.e. fewer numbers with bigger fonts).
5) Fig.1. The lake acronyms need to be specifically defined in the caption. Which lake is which? Only four lakes are shown but six lakes appear to be represented in Fig. 2. Explain why in the caption of Fig.2. The scales and UTM coordinates are illegible.
6) Fig. 4. The y scale needs to be simplified with fewer values reported with fewer significant digits. The initial Y value appears to be reported as zero, which seems incorrect considering the rest of the scale. If these are box and whisker plots need to define in the caption what is being plotted; Median? Range, 25 and 75th percentiles? Values excluded?
7) Fig. 5; most of the text is illegible; font too small
8) Tables 2 and 3; to save space, report correlation values to two significant digits, not three.
9) Fig.6; This figure will be of little use to most of the readers because the caption is inadequately descriptive; define the acronyms; what do the values on the x and y axes represent? What do the red and blue colours represent? Tag the a) and b) panels directly with the lake names. Most of the text is barely legible. Again, the caption must allow the figure to stand alone, without the necessity of referring to the text.
L25; name the Wuhan lakes
L25-26; state which lake is more eutrophic
L27; define what nrfA gene is
L39; this statement may be considered controversial because P is normally considered to be the limiting nutrient for algal growth in freshwater lakes, not N.
L67; ‘Song’; provide full reference.
L103; how can lake be 800 km2 in area but only 0.98m deep?
L158; Lake CHW does not appear to have higher TP as stated (from Fig. 2).
L259; ‘ The concentrations of DNRA bacteria were higher than that’ of other bacteria that effect N. This finding is important but it is unclear which figure actually shows this relationship. Please be more specific, referring to the appropriate figure and preferably give some values here as well.
I hope these comments assist in improving the manuscript.
Author Response
Response to Reviewer 1 Comments
Reviewed Dec. 17, 2019
This paper presents an extensive data set relating the bacterial communities and environmental conditions in four Chinese lakes with varying degrees of eutrophication. An important finding is the relationship between NO3, NH4, DNRA bacteria and correlation with cyanobacteria outbreaks in one of the lakes (CHW). Although, I’m not an expert in microbiology, this appears to be a significant finding and this, along with the other correlations presented particularly in Tables 2 and 3, I feel, makes this study worthy of publication. I have included a number of editing comments below, aimed at improving the readability of the manuscript;
Point 1: 1) Table 1; The caption is incomplete. Which lake is which? Are these surface readings, or at depth, or interstitial water? Were they taken at different locations on the same day or different times at the same location? This information needs to be included in the caption to allow the figure to ‘stand alone’ without the necessity of referring to the text.
2)Table 1; Generally too many significant digits reported; e.g. report DO to only one decimal and don’t report decimals for ORP. Please check ORP values. They appear too low (<200 mV) for aerobic water. Perhaps forgot to add the probe constant (~200 mV) to the field reading.
3)Table 1; All of the readings (3-8) from each lake are generally similar. Thus, there is no need to report all these numbers. Simply report the mean +/- sd and n for each lake to save space.
Response 1: Thanks for your good suggestions. We have removed Table 1 and put it in text in L156-158.
Point 2: 4) With the space saved by condensing Table 1, double the size of Fig. 2. This figure is an important figure but it is essentially illegible because the font size is much too small, etc. Decrease the number of values on the Y axes and increase their font size. e.g. list TN values at 0, 5 and 10 only, not 0, 2, 4, 6, 8, 10. Simplify the Y values on all the plots similarly. This is particularly important because you have used different Y scales for each of the N and P parameters, which at first glance, makes the values look similar when actually they are much different. The reader needs to see immediately that the scales are different (i.e. fewer numbers with bigger fonts).
5) Fig.1. The lake acronyms need to be specifically defined in the caption. Which lake is which? Only four lakes are shown but six lakes appear to be represented in Fig. 2. Explain why in the caption of Fig.2. The scales and UTM coordinates are illegible.
6) Fig. 4. The y scale needs to be simplified with fewer values reported with fewer significant digits. The initial Y value appears to be reported as zero, which seems incorrect considering the rest of the scale. If these are box and whisker plots need to define in the caption what is being plotted; Median? Range, 25 and 75th percentiles? Values excluded?
7) Fig. 5; most of the text is illegible; font too small
8)Tables 2 and 3; to save space, report correlation values to two significant digits, not three.
9) Fig.6; This figure will be of little use to most of the readers because the caption is inadequately descriptive; define the acronyms; what do the values on the x and y axes represent? What do the red and blue colours represent? Tag the a) and b) panels directly with the lake names. Most of the text is barely legible. Again, the caption must allow the figure to stand alone, without the necessity of referring to the text.
Response 2: Thanks for your good suggestions and valuable comments. We have modified all the figures and tables according to your request.
Point 3: L25; name the Wuhan lakes
L25-26; state which lake is more eutrophic
L27; define what nrfA gene is
Response 3: Thanks for your good suggestions. The relative statements have been revised in L25-28.
Point 4: L39; this statement may be considered controversial because P is normally considered to be the limiting nutrient for algal growth in freshwater lakes, not N.
Response 4: Thanks for your good suggestions. Here we want to express that N is only one of the key limiting elements for lake eutrophication. Indeed, P is also a very important factor. However, we focus on the N cycle, so just briefly introduce it at the beginning but not overstate its importance.
Point 5: L67; ‘Song’; provide full reference.
L103; how can lake be 800 km2 in area but only 0.98m deep?
Response 5: Thanks for your reminder. The relative reference and statement have been revised in L68 and L104, respectively.
Point 6: L158; Lake CHW does not appear to have higher TP as stated (from Fig. 2).
Response 6: Thanks for your good suggestions. There is little difference between TP of the east and west basins of Lake Chaohu, and the nutrient difference is mainly reflected in the distribution of N and chlorophyll a.
Point 7: L259; ‘The concentrations of DNRA bacteria were higher than that’ of other bacteria that effect N. This finding is important but it is unclear which figure actually shows this relationship. Please be more specific, referring to the appropriate figure and preferably give some values here as well.
Response 7: Thanks for your good suggestions and valuable comments. This is based on the references of other researchers and the relative discussion has been added in L280-281.

Reviewer 2 Report
The authors conducted molecular biological tools for investigating the community abundance of DNRA bacteria and its reduction function in shallow eutrophic lakes. For this objective, they performed qPCR, cloning, and sequencing which allowed us to analyze the nrfA gene biodiversity and the change of community under different trophic levels. The authors concluded that the nrfA gene abundance is useful to assess the DNAR process in biological activity lakes. From this view point, the manuscript represents an interesting and useful input. However, I think that it is several flaws that must be faced before being usable for publication.
One of the these is that the manuscript should provide the details of data collection. Line 108-110, please express the date, and depth when the surface water(e.g. DO, Chl a, nutrients) and sediment samples were collected. Line 184, Table 1 showed us that had to present the specific different trophic levels related to transparency. The main concern is that those data did not provide critical information supporting your purpose. Line 251, Figure6(a), “NrfA” should be “nrfA”. In discussion section, Line 255-265, authors must provide more faithful representation regarding the measurement of nrfA gene abundance and its environmental factors using this MS case study Lakes (For example, Lake Chaohu and Lake Qingling).Author Response
Response to Reviewer 2 Comments
The authors conducted molecular biological tools for investigating the community abundance of DNRA bacteria and its reduction function in shallow eutrophic lakes. For this objective, they performed qPCR, cloning, and sequencing which allowed us to analyze the nrfA gene biodiversity and the change of community under different trophic levels. The authors concluded that the nrfA gene abundance is useful to assess the DNAR process in biological activity lakes. From this view point, the manuscript represents an interesting and useful input. However, I think that it is several flaws that must be faced before being usable for publication.
One of the these is that the manuscript should provide the details of data collection.
Point 1: Line 108-110, please express the date, and depth when the surface water(e.g. DO, Chl a, nutrients) and sediment samples were collected.
Response 1: Thanks for your good suggestions. The sampling date has been mentioned in L90. In addition, the collection depths for water and sediment samples have been added in L109-110.
Point 2: Line 184, Table 1 showed us that had to present the specific different trophic levels related to transparency. The main concern is that those data did not provide critical information supporting your purpose.
Line 251, Figure6(a), “NrfA” should be “nrfA”.
Response 2: Thanks for your good suggestions. We have removed Table 1 and put it in text in L156-158 and modified the figures.
Point 3: In discussion section, Line 255-265, authors must provide more faithful representation regarding the measurement of nrfA gene abundance and its environmental factors using this manuscript case study Lakes (For example, Lake Chaohu and Lake Qingling).
Response 3: Thanks for your good suggestions and valuable comments. The measurement of nrfA gene abundance and environmental have been mentioned in 2. Materials and Methods section L114-146. The same primers and qPCR techniques were used in other references cited, so this comparison is of reference value which have been added in L278-282. In addition, the relationship between gene abundance and environmental factors is mentioned in the Discussion section L287-324.
